

**AnisoVeg: Anisotropy and Nadir-normalized MODIS MAIAC datasets for satellite**
**vegetation studies in South America**
Ricardo Dalagnol[1,2,3,*], Lênio Soares Galvão[3], Fabien Hubert Wagner[1,2], Yhasmin Mendes
Moura[4,5], Nathan Gonçalves[6], Yujie Wang[7,8], Alexei Lyapustin[7], Yan Yang[1], Sassan Saatchi[1,2],
Luiz Eduardo Oliveira Cruz Aragão[3,9]
Center for Tropical Research, Institute of the Environment and Sustainability, University of
California, Los Angeles, Los Angeles, CA 90095, USA.
NASA-Jet Propulsion Laboratory, California Institute of Technology, Pasadena, CA 91109,
USA.
Earth Observation and Geoinformatics Division, National Institute for Space Research-INPE,
São José dos Campos, SP, 12227-010, Brazil.
Institute of Geography and Geoecology, Karlsruhe Institute of Technology, Karlsruhe,
Germany.
Centre for Landscape and Climate Research, School of Geography, Geology, and the
Environment, University of Leicester, Leicester, UK.
Michigan State University, Department of Forestry, College of Agriculture & Natural
Resources, East Lansing, MI, USA.
NASA Goddard Space Flight Center, Greenbelt, MD, United States.
Joint Center for Earth Systems Technology, University of Maryland Baltimore County, 1000
Hilltop Circle, Baltimore, MD.
Geography, College of Life and Environmental Sciences, University of Exeter, Exeter EX44RJ,
UK.

* Corresponding author. E-mail: ricds@hotmail.com.

**This document includes:** main manuscript, figures and tables.





**Abstract**

The AnisoVeg product consists of monthly 1-km composites of anisotropy (ANI) and nadir-normalized (NAD) surface reflectance layers obtained from the Moderate Resolution Imaging Spectroradiometer (MODIS) sensor over the entire South America. The satellite data were pre-processed using the Multi-Angle Implementation Atmospheric Correction (MAIAC). The AnisoVeg product spans 22 years of observations (2000 to 2021) and includes the reflectance of MODIS bands 1 to 8 and two vegetation indices (VIs): Normalized Difference Vegetation Index (NDVI) and Enhanced Vegetation Index (EVI). While the NAD layers reduce the data variability added by bidirectional effects on the reflectance and VI time series, the unique ANI layers allow the use of this multi-angular data variability as a source of information for vegetation studies. The AnisoVeg product has been generated using daily MODIS MAIAC data from both Terra and Aqua satellites, normalized for a fixed solar zenith angle (SZA = 45º), modelled for three sensor view directions (nadir, forward, and backward scattering), and aggregated to monthly composites. The anisotropy was calculated by the subtraction of modelled backward and forward scattering surface reflectance. The release of the ANI data for open usage is novel, as well as the NAD data at an advance processing level. We demonstrate the use of such data for vegetation studies using three types of forests in eastern Amazon with distinct gradients of vegetation structure and aboveground biomass (AGB). The gradient of AGB was positively associated with ANI, while NAD values were related to different canopy structural characteristics. This was further illustrated by the strong and significant relationship between $EVI_{ANI}$ and forest height observations from the Global Ecosystem Dynamics Investigation (GEDI) LiDAR sensor considering a simple linear model ($R^2 = 0.55$). Overall, the time series of the AnisoVeg product (NAD and ANI) provide distinct information for various applications aiming at understanding vegetation structure, dynamics, and disturbance patterns. All data, processing codes and results are made publicly available to enable research and the extension of AnisoVeg products for other regions outside the South America. The code can be found at https://doi.org/10.5281/zenodo.6561351 (Dalagnol and Wagner, 2022), $EVI_{ANI}$ and $EVI_{NAD}$ can be found as assets in the Google Earth Engine (GEE) (described in the data availability section), and the full dataset is available at the open repository <https://doi.org/10.5281/zenodo.3878879> (Dalagnol et al., 2022).

**Key-words:** AnisoVeg, South America, vegetation structure, forest monitoring, MODIS.

## 1. Introduction

The anisotropy is defined by the directional dependence of observations on mechanical or physical properties of surfaces. Because most land covers are not Lambertian (isotropic), the surface reflectance measured by satellite sensors varies with the view zenith angle (VZA), view direction (backward or forward scattering), and solar zenith angle (SZA) (Galvão et al., 2011). This is especially valid for images acquired over vegetated surfaces by large field-of-view (FOV) instruments such as the Moderate Resolution Imaging Spectroradiometer (MODIS) (Bhandari et al., 2011). MODIS has a wide swath scanning ±55º from nadir on board the Terra and Aqua satellites. For example, a reflected signal coming from the backward scattering direction of MODIS under a large VZA and close-to-zero relative azimuth angle (RAA) between the satellite and sun (sun behind the platform) is generally higher than that coming from the nadir (VZA = 0º) or forward scattering direction (platform facing the sun at RAA = 180º). Moreover, the SZA also varies seasonally and across geographical locations, affecting the amounts of shadows in the surfaces observed by satellites (Galvão et al., 2013). Such view-illumination effects are dependent on the land cover types and their magnitude relates to differences in biophysical properties of the vegetation (Foody & Curran, 1994). Therefore, the vegetation anisotropy can be seen antagonistically as sources of noise and biophysical information in the time-series analysis of



vegetation indices (VIs) calculated from MODIS. As a source of noise, one may consider that the
reflected signal toward the large FOV satellite sensors varies with distinct view-illumination
geometries of data acquisition over the same surface. As a source of information, one may
highlight that the anisotropy is land-cover type dependent, showing spectral variations that may
be associated, for instance, with changes in vegetation structure across different forests.
To reduce the bidirectional effects as a source of noise, a nadir-normalized dataset can be created.
We can normalize the surface reflectance of the MODIS bands to a specific set of VZA and SZA
using the bidirectional reflectance distribution function (BRDF), represented by a model such as
the Ross-Thick Li-Sparse (RTLS) (Wanner et al., 1995). To ensure confidence in the data
analysis, we can also use the Multi-Angle Implementation Atmospheric Correction (MAIAC) for
atmospheric correction. MAIAC is a new generation of cloud screening and atmospheric
correction algorithm that uses an adaptive time series analysis and processing of groups of pixels
to derive atmospheric aerosol concentration, cloud mask and surface reflectance without typical
empirical assumptions (Lyapustin et al., 2011, 2012). It offers substantial improvement over
conventional algorithms by mitigating atmospheric interference and advancing the accuracy of
surface reflectance over tropical vegetation by a factor of 3 to 10 (Hilker et al., 2012). Due to the
improvements in cloud detection, aerosol retrieval and atmospheric correction, the MAIAC
algorithm provides from 4 to 25% more high-quality retrievals than the traditional MOD09
product, with the largest estimate being observed for tropical regions (Lyapustin et al., 2021).
Studies have used MODIS MAIAC observations with nadir-normalized geometry to assess
Amazonian forests' structure, functioning, and impacts of environmental and climate change
(Hilker et al., 2014; Wagner et al., 2017; Anderson et al., 2018; Dalagnol et al., 2018; Fonseca et
al., 2019; Bontempo et al., 2020; Gonçalves et al., 2020; Zhang et al., 2021). For instance, such
product provided reliable time series of surface reflectance data that allowed to identify large-
scale communities of bamboo species and their dynamics in the southwest Amazon (Dalagnol et
al., 2018). Lastly, by improving the cloud screening and minimizing BRDF artifacts in
comparison to uncorrected data, the MAIAC greatly contributed to the understanding of the long-
standing debate in the Amazon over the possible existence of the green-up phenomenon observed
during the dry season of each year or with severe droughts (Morton et al., 2014; Bi et al., 2015;
Saleska et al., 2016; Wu et al., 2017). The existence of this phenomenon has implications on the
comprehension of the resilience of tropical forests to climate change.
To use the bidirectional effects as a source of information, we generate an anisotropy dataset that
is dependent on land-cover types and captures the variations of sunlit and shaded canopy
components viewed by the sensors (Chen et al., 2003; Gao, 2003). The use of multi-angular
information to obtain metrics of anisotropy and extract information on forest structure was
suggested two decades ago (Foody & Curran, 1994). The first experiments with such concept
were conducted by calculating the ratio between backward and forward scattering data and
generating the anisotropy index (ANIX) on studying short-stature grass-type vegetation
(Sandmeier et al., 1998). Other indices have been developed and validated afterwards (Schaaf et
al., 2002; Lacaze et al., 2002; Chen et al., 2005; Pocewicz et al., 2007; Moura et al., 2015; Sharma
et al., 2021). However, this remains an understudied topic with limited results reported in the
literature, especially in tropical regions. For instance, observations from the Multi-angle Imaging
Spectroradiometer (MISR)/Terra in the backward and forward scattering directions facilitated the
discrimination of savanna physiognomies in Brazil (Liesenberg et al., 2007). MODIS MAIAC
data from both directions were also used to calculate an anisotropic VI that explained part of the
large-scale photosynthetic activity in the Amazon, where higher photosynthetic activity was
associated to higher anisotropy values (Sousa et al., 2017). Moura et al. (2015) employed a more
sophisticated approach based on scattering at backward and forward view directions using multi-
temporal and multi-angular observations of MAIAC MODIS and BRDF modelling. The resultant
metrics of anisotropy were further validated against field and airborne Light Detection And



Ranging (LiDAR) observations, showing strong linear relationship with leaf area index (LAI) ($R^2$
= 0.70-0.88), canopy heterogeneity ($R^2$ = 0.54), and photosynthetic activity ($R^2$ = 0.73-0.98)
(Moura et al., 2015; Moura et al., 2016; Hilker et al., 2017). Although showing great potential in
vegetation studies, the aforementioned anisotropy metrics were never computed over larger areas
of the world such as proposed in this study for South America.
The objective of this work is to present the AnisoVeg product, and how it can be used for
vegetation studies. We use MODIS Collection 6 (C6) MAIAC (Lyapustin et al., 2018) monthly
data (2000-2021) generated at 1-km spatial resolution for the entire South America with two
different types of layers: (1) nadir-normalized (NAD) data for the surface reflectance of MODIS
bands 1 to 8 and two VIs (NDVI and EVI); and (2) anisotropy data (ANI) calculated from the
difference between backward and forwarding scattering estimates of bands 1 to 8 and VIs (Moura
et al., 2015). The motivations for generating this product extend from developing applications of
multi-angle observations for vegetation studies to producing analysis-ready and openly available
datasets of anisotropy and nadir metrics for a larger community of users. The paper is organized
in several sections to present the processing steps for generating the AnisoVeg products, a brief
evaluation of data products over experimental areas, and finally an example of its potential
application in vegetation studies.

## 2. Methodology to compute the AnisoVeg product

### 2.1. Daily MODIS MAIAC surface reflectance data over South America

Daily surface reflectance data were obtained from the MODIS product MCD19A1 v006
(collection 6) for the tiles covering South America (Figure 1). According to the MODIS traditional
tiling system, these tiles ranged from 9-14 (horizontal) and 7-14 (vertical). The input data
consisted in cross-calibrated surface reflectance from Terra and Aqua satellites on eight spectral
bands (Table 1) with 1-km spatial resolution from 2000 to 2021 (Lyapustin & Wang, 2018;
http://dx.doi.org/10.5067/MODIS/MCD19A1.006). This product provides surface reflectance
data corrected for atmospheric effects by the MAIAC algorithm, and controlled for cloud-free
and clear-to-moderately turbid conditions with Aerosol Optical Depth (AOD) at 0.47 µm below
1.5 (Lyapustin et al., 2018). The raw data were obtained from the NASA's Level-1 and
Atmosphere Archive and Distribution System (LAADS) Distributed Active Archive Center
(DAAC) available at https://ladsweb.modaps.eosdis.nasa.gov/archive/allData/6/MCD19A1/.

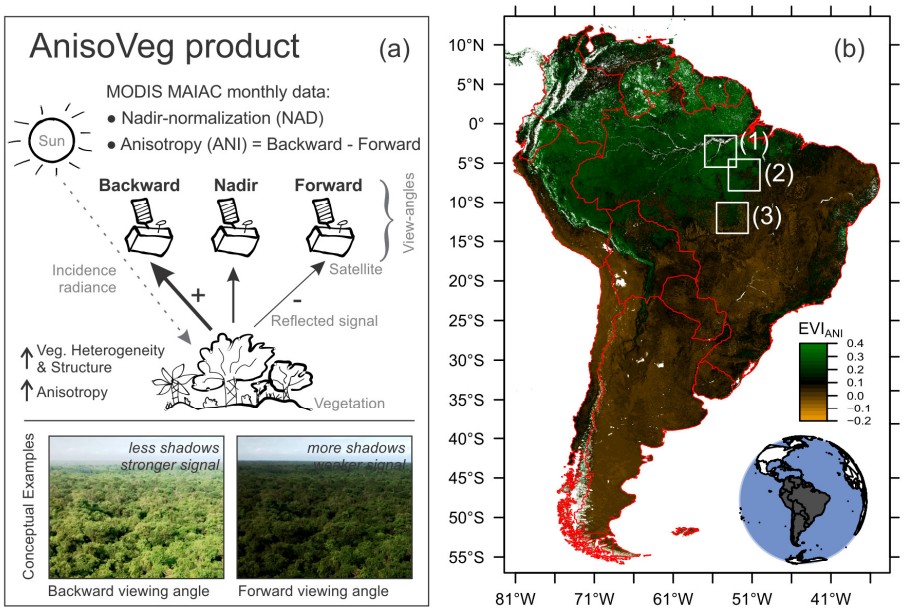

Figure 1 – AnisoVeg product concept and the area of coverage. (a) Schematic representation showing the observational geometry and the processing steps for producing NAD and ANI data from MODIS and to provide information on vegetation heterogeneity and structure, and (b) the visualization of the anisotropy EVI ($EVI_{ANI}$) for South America from August 2021 at 1-km spatial resolution, showing the coverage of the product in South America and the location of three sites used to demonstrate potential applications. The sites are: (1) Tapajós National Forest, (2) São Felix do Xingu, and (3) Xingu Park. Red lines indicate the countries boundaries.

Table 1 – MODIS spectral bands. NIR = near infrared; SWIR = shortwave infrared.

| Band number | Band name | Wavelength (nm) |
|---|---|---|
| 1 | Red | 620–670 |
| 2 | NIR-1 | 841–876 |
| 3 | Blue-1 | 459–479 |
| 4 | Green | 545–565 |
| 5 | NIR-2 | 1230–1250 |
| 6 | SWIR-1 | 1628–1652 |
| 7 | SWIR-2 | 2105–2155 |
| 8 | Blue-2 | 405–420 |

## 2.2. The AnisoVeg product

The AnisoVeg product consists of two types of data spanning from 2000 to 2021 in monthly composites at 1-km spatial resolution: (a) the nadir-normalized (NAD) data; and (b) the anisotropy (ANI) data. Each data type has 10 layers corresponding to the MODIS bands 1 to 8, and two VIs (NDVI and EVI).



### 2.2.1. The nadir-normalized (NAD) data

In order to minimize the differences in sun-sensor geometry between the MODIS scenes and generate the NAD dataset, the daily surface reflectance data were normalized to a fixed 45° SZA and to nadir observation (VZA = 0°) using the BRDF and the Ross-Thick Li-Sparse (RTLS) model (Lucht and Lewis, 2000). Parameters of the RTLS BRDF model are part of the MAIAC product suite (MCD19A3 product) reported every 8 days. The closest RTLS parameters in time were used to normalize the daily data. The normalized Bidirectional Reflectance Factor ($BRFn$) for the NAD surface reflectance (SZA = 45°, VZA = 0°, RAA = 0°) was calculated using Eq. 1 (Lyapustin et al., 2018):

$$BRFn \ = \ BRF \ \times \ \frac{k^L + F_{0V} \times k^V + F_{0G} \times k^G}{k^L + F_V \times k^V + F_G \times k^G} \tag{1}$$

where $k^L$, $k^V$, and $k^G$ are the BRDF isotropic, volumetric, and geometric-optical kernel weights, respectively; $F_{0V}$ and $F_{0G}$ are the BRDF kernel values for the given geometry listed in Table 2; and $F_V$ and $F_G$ are the kernel values of the RTLS model for the specific MODIS observation, respectively (Lyapustin et al., 2018). $F_V$ and $F_G$ values are available at 5-km cells and were resampled to 1-km using the nearest neighbors' method to match the spatial resolution of the spectral bands. This resampling step does not create spatial artifacts in the data because the geometry changes slowly over time (Lyapustin et al., 2018).

Table 2 – View-angle normalizations and corresponding BRDF kernel values.

| View-angle | $F_{0V}$ | $F_{0G}$ |
|---|---|---|
| **Nadir** | -0.04578 | - 1.10003 |
| **Backward scattering** | 0.22930469 | 0.017440045 |
| **Forward scattering** | -0.12029795 | -1.6218740 |

We aggregated normalized daily data into monthly composites by keeping the median values for each pixel. During the temporal aggregation, we also calculated the per-pixel number of samples (or observations) for each monthly composite, which can be used as auxiliary data to filter pixels with low number of observations (less reliable estimates of surface reflectance). The tiles were mosaicked for the entire South America and then re-projected from the original sinusoidal projection to the geographic coordinates system (datum WGS-84, EPSG 4326). The output spatial resolution corresponded to 0.009107388 degrees, which is approximately equivalent to 1 km in projected coordinates.

We also calculated two traditional vegetation indices: NDVI (Rouse et al., 1973) (Eq. 2) and EVI (Huete et al., 2002) (Eq. 3).

$$NDVI \ = \ \frac{\rho NIR - \rho Red}{\rho NIR + \rho Red} \tag{2}$$

$$EVI \ = \ 2.5 \ * \ \frac{\rho NIR - \rho Red}{\rho NIR + (6 * \rho Red - 7.5 * \rho Blue) + 1} \tag{3}$$

where $\rho$ is the surface reflectance of a MODIS band, $\rho NIR$ is the NIR reflectance (band 2), $\rho Red$ is the red reflectance (band 1), and $\rho Blue$ is the blue reflectance (band 3). The constants in Eq. 3 (6, 7.5, 1, and 2.5) represent: the aerosol coefficient adjustment of the atmosphere for the red and blue bands; the adjustment factor for the soil; and the gain factor, respectively (Huete et al., 2002).





### 2.2.2. The anisotropy (ANI) data

For the ANI data, the daily surface reflectance data was first normalized to two viewing-angles at the backward (SZA = 45º, VZA = 35º, RAA = 180º) and forward (SZA = 45º, VZA = 35º, RAA = 0º) scattering using Eq. 1 and values from Table 2. To minimize potential errors of BRDF extrapolation, the VZA was set to 35º instead of the hotspot (45º), because 35º is a very common VZA in the empirical data distribution of the South America, and thus providing better estimates of the anisotropy (Moura et al., 2015). The standard deviation for this modelling was thoroughly investigated in a previous study and determined as 10% of the observed variation in anisotropy (Moura et al., 2015). Further, we aggregated the backward and forward scattering data temporally into monthly composites following the same procedures as before for the NAD data. We then calculated the NDVI and EVI for each of the view-angle normalizations. Finally, we obtained the difference between backward and forward scattering estimates for each of the eight MODIS bands, as well as for the NDVI and EVI, effectively generating the ANI layers (Eq. 4; Moura et al., 2015):

$$ANI_i = Backward_i - Forward_i \qquad (4)$$

where $i$ is the spectral band or VI selected in the calculation.

### 2.3. Algorithm and computation

All data processing was done in R v4.0.2 (R Core Team, 2016) and the code is available at GitHub (https://github.com/ricds/maiac_processing) (Dalagnol & Wagner, 2022). Besides processing the AnisoVeg product from the daily MAIAC MODIS data, the code can also generate 16-day or 8-day temporal composites, mosaics, and VIs. Although we focused on South America when developing AnisoVeg, the code can readily be adapted to process data for other parts of the world and generate corresponding NAD and ANI layers. Below, we provide the computer specification for anyone who wishes to process the data independently.

For the presented dataset, the computation was performed under a HP Z840 Workstation with Intel Xeon CPU E5-2640 v3 (2.60Ghz, 32 cores), and 64 Gb RAM memory. The daily MODIS data for the whole South America from 2000 to 2021 accounted for 6.69 Tb. Processing monthly composites is computationally intensive due to loading all daily data for each month at once for a given tile. Thus, the main bottlenecks are RAM memory and hard drive writing speed. For the workstation with 64 Gb memory, the usage of 10 cores running in parallel processing was the optimal choice. The average processing time of each monthly composite for one tile was 6 minutes. Therefore, it took 26.2 hours to process the 262 composites (March 2000 to December 2021) for each tile. Since we had 31 tiles covering the South America, the total amount of time to process one view-normalization was approximately a month (33.8 days). Consequently, the total time spent in computation was 101.5 days for processing the three view-normalizations (nadir, backward, and forward scattering) and generating the NAD and ANI layers. Processing can also be done with less potent computers with a minimum of 16 Gb RAM memory and 4 processing cores.

### 2.4. Time series availability and uncertainty

The monthly compositing process returned a time series dataset over all of South America with an average of 242 ± 35 out of a maximum of 262 composites (period between March 2000 and December 2021) for each pixel with some data missing due to lack of high-quality observations (Figure 2). Only 34.3% of the available pixels have the full time series (262 composites). The



Amazon region shows a lower mean number of samples in the time series with an average of 231
± 29 composites, which can be seen in Figure 2. This lower number of samples is due to the innate
high cloud cover (Durieux et al., 2003). It is important to note that the AnisoVeg product was
strictly created to analyze land surface and does not cover water bodies. Moreover, the period
between March 2000 and June 2002 has higher amounts of missing data because it preceded the
launch of the Aqua satellite. When data from both satellites (Terra and Aqua) were combined to
create the product after 2002, we had a much better pixel level data availability to produce dense
time series. Although we have a dense time series across the Amazon rainforests (Figure 2a), the
mean number of daily observations within a month for this region is relatively lower than that
observed in more dry and seasonal regions of South America (Figure 2b). Thus, we suggest using
the number of samples layer as a proxy for uncertainty on the retrieval of monthly composites to
filter out pixels with low number of samples (e.g., less than three observations per composite).
The lesser number of samples one pixel has, the higher the uncertainty in the data analysis.

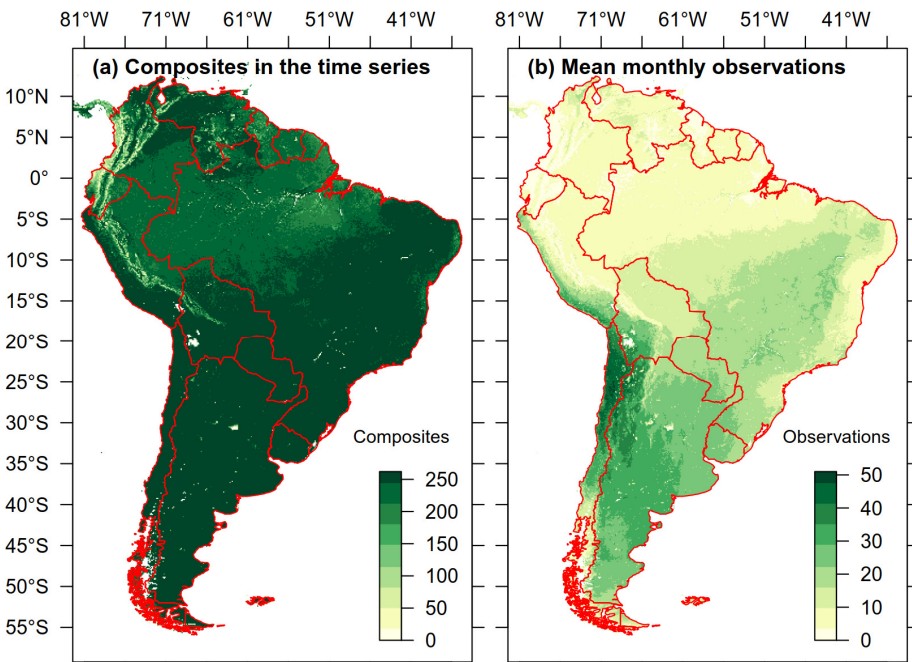


Figure 2 – AnisoVeg time series availability and uncertainty over South America. (a) The number
of composites in the time series representing pixel availability. The maximum number of
composites in the time series is 262 for the period between March 2000 and December 2021. (b)
Mean number of daily observations within a month used to create the monthly composites as a
proxy for uncertainty. The maximum daily observations in a composite are 60 (twice a day every
day for a month).

**3. Spatial and temporal distribution of NAD and ANI data across the Amazon forests**
To demonstrate the spatial and temporal distribution of NAD and ANI data over the Brazilian
Amazon rainforests, we selected three experimental areas (rectangles in Figure 1). These areas
show old-growth rainforests with distinct canopy structure and aboveground biomass (AGB)
stocks. The AGB increases from semideciduous forests at the Xingu Park (190 ± 19 Mg ha$^{-1}$) and
open ombrophilous forests with lianas at the São Felix do Xingu ($241 \pm 31$ Mg ha$^{-1}$) to dense
ombrophilous forests at the Tapajós National Forest ($288 \pm 38$ Mg ha$^{-1}$), as estimated by the
ESA/CCI AGB map from 2017 (Santoro & Cartus, 2021). These are large-scale AGB estimates
and may underestimate the true AGB at higher values such as in the Tapajós site. These three
sites are also expected to show different phenological dynamics because their selected pixels
cover distinct phenoregions in the study reported by Xu et al. (2015).
When compared to the nadir-normalized EVI ($EVI_{NAD}$) images (Figures 3a, b, c), the anisotropy
EVI ($EVI_{ANI}$) data showed different spatial patterns across sites (Figures 3d, e, f). While the
forests over the three sites showed approximately similar $EVI_{NAD}$ values ($EVI_{NAD} \approx 0.50$) (Figures
3a,b,c), they showed more variability in $EVI_{ANI}$ between the Xingu Park ($EVI_{ANI} > 0.20$), São
Felix do Xingu ($EVI_{ANI} > 0.24$), and Tapajós ($EVI_{ANI} > 0.27$) sites (Figures 3d,e,f). This increase
in $EVI_{ANI}$ between sites goes into the same direction of the AGB gradient observed from the
Xingu Park to the Tapajós National Forest. This result may indicate different forest canopy
structures that were not captured in the $EVI_{NAD}$ observations, but were captured by the $EVI_{ANI}$.
Overall, the $EVI_{ANI}$ is high over forests (0.20 to 0.30) and low over pastures and crops (less than
0.10). This means large anisotropy between the reflected energy in backward and forward
scattering MODIS directions due to the structural complexity of forest canopies. The association
between anisotropy and forest canopy structure has been previously shown for the same region in
a previous work (Moura et al., 2016).

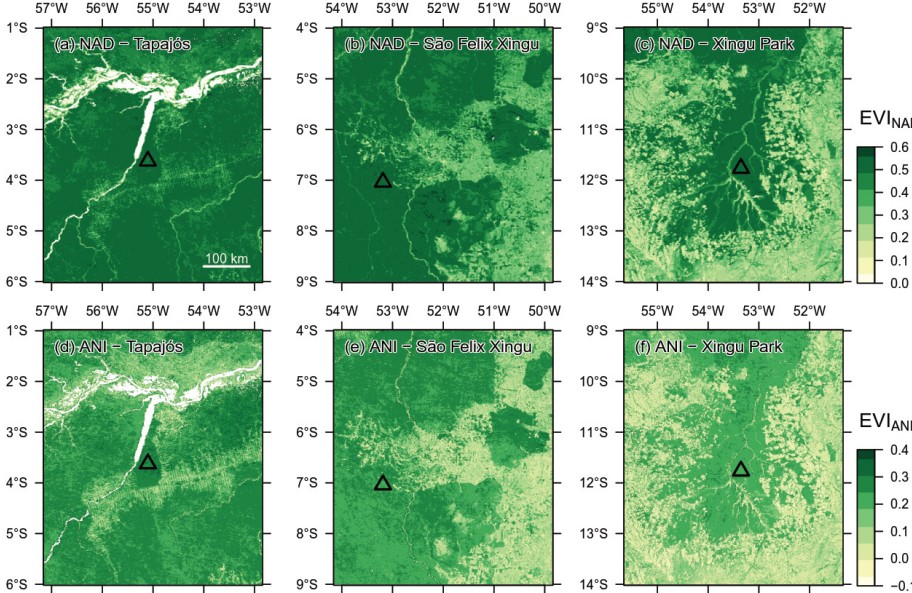


Figure 3 – The spatial distribution in August 2020 (dry season) of the nadir-normalized Enhanced
Vegetation Index ($EVI_{NAD}$) is shown in (a), (b), and (c) for the Tapajós National Forest, São Felix
do Xingu and Xingu Park, respectively. Corresponding results for the anisotropy EVI ($EVI_{ANI}$)
are shown in (d), (e), and (f), respectively. The triangles plotted over (a, b, and c) indicate the
sites used to obtain the profiles of Figure 4.
From the comparison of different sites (triangles in Figure 3a), we observed that the mean $EVI_{NAD}$
signal over the time period did not vary much between the selected forests, while the $EVI_{ANI}$
varied greatly (Figure 4): Tapajós (mean $EVI_{NAD} = 0.49$, mean $EVI_{ANI} = 0.27$), São Felix do Xingu
(mean $EVI_{NAD} = 0.51$, mean $EVI_{ANI} = 0.24$), and Xingu Park (mean $EVI_{NAD} = 0.51$, mean $EVI_{ANI}$



= 0.22). Moreover, EVI$_{NAD}$ and EVI$_{ANI}$ values were moderately positively correlated at Tapajós
($r$ = +0.37), weakly correlated at São Felix do Xingu ($r$ = +0.06), and moderately negatively
correlated at the Xingu Park ($r$ = -0.28). The EVI$_{NAD}$ and EVI$_{ANI}$ are seasonal variability and phase
correlation changes from site to site, suggesting that different canopy dynamics processes are
likely being captured by the two metrics at the three sites. Understanding exactly what those
effects mean for these forests is beyond the scope of this paper. However, it indicates open venues
for studying forest functioning using these products. For example, previous studies have shown
that EVI$_{NAD}$ metrics captured different compositions of leaf ages in the canopies of central
Amazon (Gonçalves et al., 2020).

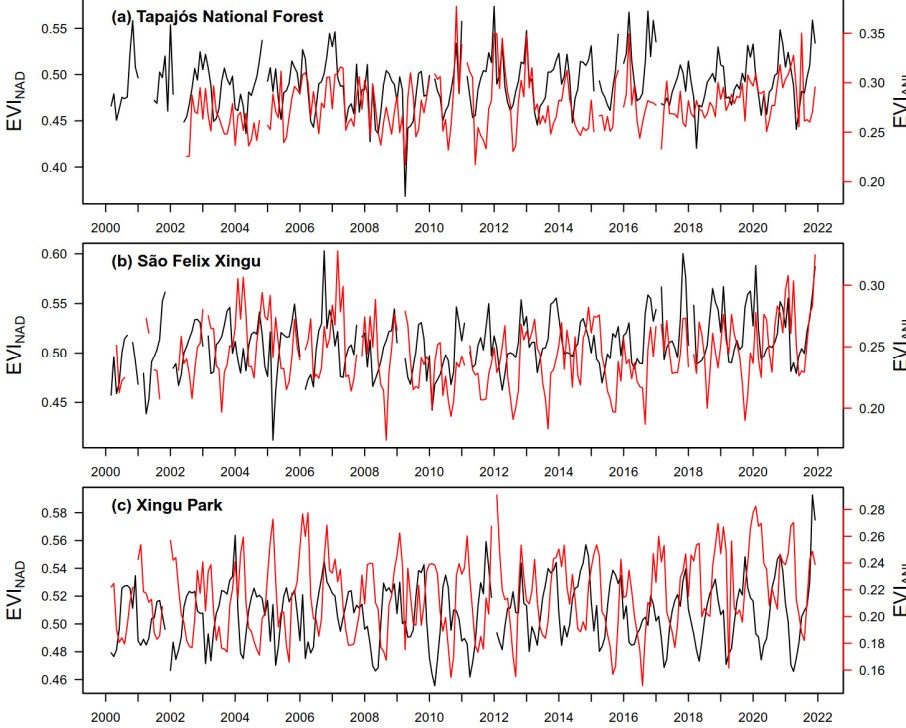


Figure 4 – Time series of AnisoVeg's MODIS Enhanced Vegetation Index (EVI) from 2000 to
2021 for old-growth forests of the (a) Tapajós National Forest; (b) São Felix do Xingu; (c) Xingu
Park. The black line indicates the nadir-normalized signal (NAD layer), while the red line
represents the EVI anisotropy (ANI layer). The profiles are the mean value of 3 x 3 pixels whose
locations are indicated by triangles in Figure 3.
To demonstrate the potential of AnisoVeg for large-scale forest structure inference, we compared
the NAD and ANI data against forest height measurements from the Global Ecosystem Dynamics
Investigation (GEDI) LiDAR sensor. We found that EVI$_{ANI}$ was able to explain up to 55% of
height variability of Amazon forests according to a simple linear relationship ($R^2$ = 0.55, $p$ < 0.01,
Figure 5). This is a very strong predicting power for a single variable, considering a simple linear
model, especially for satellite passive optical data which are often underrated for forest structure
estimates in comparison to Synthetic Aperture Radar (SAR) data. EVI$_{NAD}$ was significantly but
weakly associated to height variability ($R^2$ = 0.16, $p$ < 0.01), reinforcing the increase in
explanation power owed to the anisotropy metrics built from multi-angle observations. The height
data was derived from the GEDI LiDAR sensor aboard the International Space Station. They were

Earth System
Science
Data

obtained more specifically from the product GEDI L2A elevation and height metrics data version
2 (footprint size 25 m), acquired from April 2019 to October 2020 (available dates at the time of
download). GEDI data were downloaded from Earth Data cloud service system
(https://earthdata.nasa.gov). We selected the Relative Height metric at 98th percentile (RH98),
which represents the top canopy height. The selected RH98 metric was averaged over each 1-km
grid cell, and filtered using a threshold of greater than or equal to 50 shots per $km^2$ to have a high
confidence of reliable height estimation representing the 1-km mean. The AnisoVeg data used for
this comparison were based on the same time period as GEDI, and filtered for $EVI_{NAD}$ larger than
0.35 to exclude non-forested areas. While we only showed the plot for the strongest $EVI_{ANI}$:GEDI
relationship in June 2019 (Figure 5), the other months also showed significant ($p < 0.01$) and
strong relationships with $R^2$ ranging from 0.36 to 0.55 (mean $R^2 = 0.46$). Future studies should
explore relationships using ANI from different months and other indices, alone or in combination
with each other, to further understand their significance on explaining forest structure. This is
important to determine how the anisotropy data can contribute for aboveground biomass and
carbon estimates in conjunction with other sources of data such as those from SAR sensors.

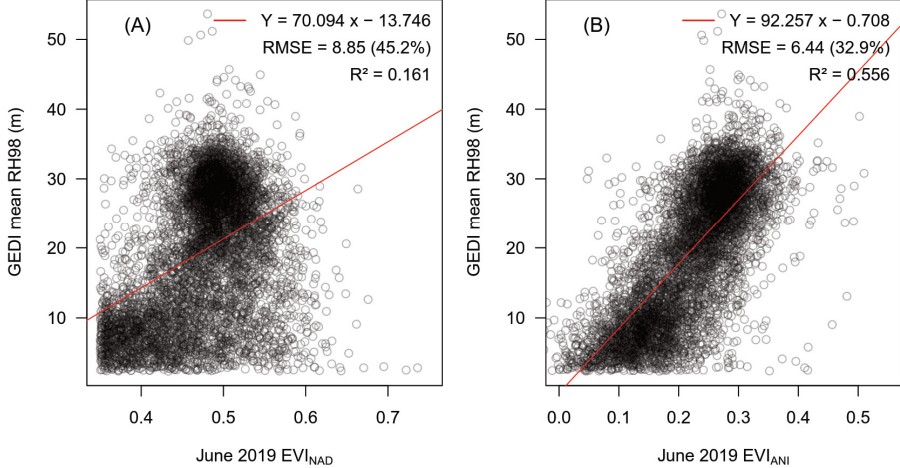


Figure 5 – Relationship between forest height (GEDI mean RH98) and two AnisoVeg layers
obtained in June 2019 over the Amazon: (a) $EVI_{NAD}$ and (b) $EVI_{ANI}$. The RH98 metric consists in
the relative height at the $98^{th}$ percentile, which represents the top of canopy height. 7,000 random
matching pixels were used in this analysis (1% of 700,000 total matching pixels available),
resulting from the filtering of both GEDI and AnisoVeg data. The red line indicates the fitted line
by a simple linear model.
In a prospective analysis, we also explored the behavior of the two EVI AnisoVeg metrics over
the Amazonian phenoregions mapped by Xu et al. (2015). The $EVI_{NAD}$ and $EVI_{ANI}$ monthly means
over different phenoregions highlighted the strong heterogeneity of the Amazonian forests
(Figure 6). For instance, the profiles showed strong differences between both metrics from
January to September in a phenoregion with well-defined dry and wet seasons (phenoregion one
in Figure 6a at the Xingu Park). Large differences between $EVI_{NAD}$ and $EVI_{ANI}$ were also observed
in some phenoregions without a very long dry season in northwest Amazon (phenoregion five in
Figure 6e). On the other hand, $EVI_{NAD}$ and $EVI_{ANI}$ showed temporal decoupling in phenoregion
three located at central-east Amazon (Figure 6c). Overall, while the seasonality of $EVI_{NAD}$ has
been investigated by many studies in the past, the seasonality of $EVI_{ANI}$ is something to be further
explored with the support of auxiliary data (e.g., airborne LiDAR and field campaigns). This is



Earth System
Science
Data

important to better understand the differences in seasonal patterns between both AnisoVeg
metrics.

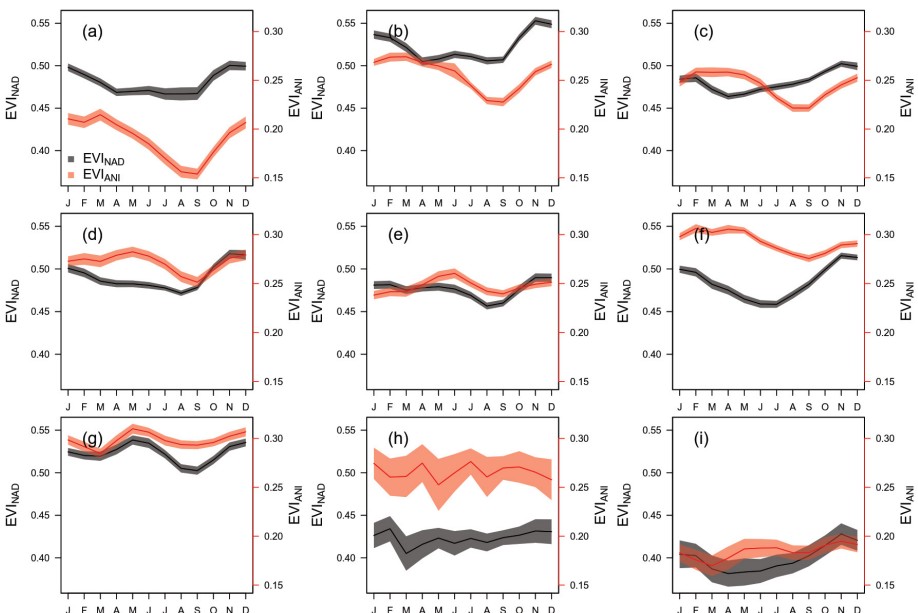


Figure 6 – Monthly means of EVI$_{NAD}$ (black) and EVI$_{ANI}$ (red) for nine phenoregions mapped by
Xu et al. (2015) in the Amazon. The phenoregions are shown in increasing order from 1 to 9 in
corresponding panels (a) to (i). They represent forests with similar seasonality and landscape
structure. Solid line and shaded area represent the mean and 95% confidence interval around the
mean. The values were extracted from 20 years of data (from 2001 to 2021) for 100 random
coordinates within each region, and extracted from 3 x 3 windows of pixels.

## 381 4. Prospective use of the dataset

The NAD layers from the AnisoVeg product have been used in previous studies to explore: the
climate drivers of the Amazon forest greening (Wagner et al., 2017); the large-scale Amazon
forest sensitivity to drought (Anderson et al., 2018); the structure and dominance of bamboo
species in southwest Amazon (Dalagnol et al., 2018); the productivity in a flooded forest in
eastern Amazon (Fonseca et al., 2019); the productivity and relationship with Sun-Induced
Fluorescence over the Brazilian Caatinga biome (Bontempo et al., 2020); the relationships with
leaf-age demography in central Amazon (Gonçalves et al., 2020); and the relationships with fire
disturbance and SAR-based Vegetation Optical Depth in southern Amazon (Zhang et al., 2021).
The ANI layers from the AnisoVeg product have been mainly used to characterize Amazon forest
structure properties (Moura et al., 2015; 2016). These layers now open new venues of
investigation on vegetation, including (but not limited to): the characterization of biophysical
attributes of forests, including their seasonality and trends; the assessment of changes in
vegetation structure due to natural disturbances or degradation (logging, fire, edge effects); and
the evaluation of forest health and productivity (greenness and browning). We expect that this
dataset contributes to upscaling studies over large areas of key forest properties such as the AGB
and canopy roughness (Foody & Curran, 1994; Saatchi et al., 2008). This information is required
for dynamic vegetation models to accurately represent the carbon cycle. This dataset is not limited



to study Amazonian forests and can be used to explore other biomes of South America such as
the Atlantic Forest, savannas (Cerrado), Caatinga, Chaco, Pantanal, and Pampas. Such studies
could improve our understanding of large-scale vegetation functioning, carbon storage, and
cycling. Ultimately, they can contribute to refine global ecosystem models, and to obtain accurate
estimates of carbon cycle in response to climate and environmental change. Furthermore,
auxiliary backward and forward scattering data are also available with the dataset. Beyond the
use of the provided ANI layers, this effectively allows the computation of several other multi-
angular anisotropy indices from the literature (Table 3), offering the possibility to investigate their
use for tropical vegetation studies.

Table 3 – Examples of other multi-angular anisotropy indices that can be further calculated using
layers of the AnisoVeg product. Lambda represents the selected spectral band or vegetation index.
N, H, and D represent nadir-view normalization, hot-spot (backward scattering), and dark-spot
(forward scattering) estimates, respectively.

| Anisotropy Indices | Formula | Reference |
|---|---|---|
| **Anisotropy index (ANIX)** | $\frac{\lambda_H}{\lambda_D}$ | Sandmeier et al. (1998) |
| **Nadir BRDF-adjusted NDVI (NDVI$_{ISO}$)** | $\frac{NIR_N - RED_N}{NIR_N + RED_N}$ | Schaaf et al. (2002) |
| **Hot-spot dark-spot index (HDS$_{RED}$)** | $\frac{RED_H - RED_D}{RED_D}$ | Lacaze et al. (2002) |
| **Normalized difference between hot-spot and dark-spot index (NDHD$_{NIR}$)** | $\frac{NIR_H - NIR_D}{NIR_H + NIR_D}$ | Chen et al. (2005) |
| **Hot-spot dark-spot NDVI (NDVI$_{HD}$)** | $\frac{NIR_H - RED_D}{NIR_H + RED_D}$ | Pocewicz et al. (2007) |
| **Hot-spot-incorporated NDVI (NDVI$_{HS}$)** | $NDVI_N \times (1 - RED_H)$ | Pocewicz et al. (2007) |
| **Anisotropy difference (ANI)\*** | $\lambda_H - \lambda_D$ | Moura et al. (2015) |
| **Vegetation Structure Index (VSI)** | $\frac{NDVI_D - NDVI_H}{1 - NIR_D}$ | Sharma et al. (2021) |

*ANI is included in the AnisoVeg product. Source: Adapted from Sharma et al. (2021).

**5. Code and data availability**
All code is available at GitHub (https://github.com/ricds/maiac_processing) (Dalagnol &
Wagner, 2022). The full dataset can be found at the official AnisoVeg repository at Zenodo
(https://doi.org/10.5281/zenodo.3878879) (Dalagnol et al., 2022). The dataset was organized in
compressed files (".zip" format) sub-divided by years (currently 2000-2021) and layers (bands 1-
8, NDVI, and EVI) for both nadir-normalization (code = NAD) and anisotropy (code = ANI). The
number of samples layers (code = NO_SAMPLES) are also provided. Inside each compressed
file there will be 12 image files (".tif" format), one per month, except for the year 2000 which
starts in March. The storage size for the whole dataset is 162.6 Gb. The data have a scale factor
of 10,000 to reduce file storage size. Thus, to obtain surface reflectance values of bands or correct
range of values for indices, you should divide the layers by 10,000. The exception is the number
of samples, which already shows the correct range of values from 0 to 60 observations. The dataset
is planned to be updated on a yearly-basis. Auxiliary data that allow the calculation of other
anisotropy metrics (listed in Table 3) are included in two separate Zenodo repositories for
backward (https://doi.org/10.5281/zenodo.6040300) (Dalagnol, 2022a) and forward scattering



(https://doi.org/10.5281/10.5281/zenodo.6048785) (Dalagnol, 2022b), including the selected
layers Red, NIR, NDVI and EVI. The $EVI_{ANI}$ and $EVI_{NAD}$ layers were also uploaded to the GEE
platform using the *geeup* tool v0.5.3 (Roy, 2022). They can be accessed through the GEE
ImageCollection assets "projects/anisoveg/assets/evi_anisotropy" and
"projects/anisoveg/assets/evi_nadir", found at
<https://code.earthengine.google.com/?asset=projects/anisoveg/assets/evi_anisotropy> and
<https://code.earthengine.google.com/?asset=projects/anisoveg/assets/evi_nadir>.

**Author contribution**
R.D. and Y.M. conceived the presented idea. R.D. designed the methodology with contributions
from Y.M. on the anisotropy method. R.D. conducted formal analysis and investigation with
contributions from L.G., F.W., N.G., and S.S. Y.W. and A.L. provided the original MODIS
(MAIAC) data and support for processing it. Y.Y. and S.S. provided the processed GEDI height
data and support to analyze it. R.D. and F.W. developed the code to process the MODIS (MAIAC)
data into the products. R.D. conducted data curation of the products. L.A. supervised the project.
R.D. wrote the original draft with support from L.G., F.W. and Y.M. All authors read, reviewed
and approved the final version of the manuscript.

**Acknowledgements**
R.D. was supported by Sao Paulo Research Foundation (FAPESP) grants 2015/22987-7 and
2019/21662-8. F.W. was supported by FAPESP grant 2015/50484-0. Part of this work was carried
out at the Jet Propulsion Laboratory, California Institute of Technology, under a contract with the
National Aeronautics and Space Administration (NASA). The funders had no role in the study
design, data collection and analysis, including the decision to publish or prepare the manuscript.
We thank the MODIS MAIAC team from NASA for providing the freely available MODIS
(MAIAC) daily dataset.

**Conflict of Interest**
The authors have declared no conflict of interest.

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
