# Peer review of "AnisoVeg: Anisotropy and Nadir-normalized MODIS MAIAC datasets for satellite"

_Earth System Science Data, 2022_

## Referee Comment (RC4)

This manuscript presents the NAD (nadir-normalized) and ANI (backward-forward) surface reflectance and VI dataset produced from MAIAC MCD19A1 daily surface reflectance and 8-day MCD19A3 product. It provides possible chances to reduce uncertainty or extract information to serve the vegetation studies. Before the publish of this manuscript, further efforts may be needed to address the following general and special comments.

General comments:

1. Several anisotropy indices except ANIX have been published in the anisotropy community. There is a need to summarize out these typical indices with advantage and disadvantage, to establish the requirement of the presented metrics NAD and ANI.

2. Confused by the "backward scattering" and "forward scattering" of Table 2, without the description of given the sun-view geometry for the fixe kernel values. Either "backward scattering" or "forward scattering" is not an sole direction.

3. Not sure why $35°$ is adopted. Is it an arbitrary determination? Through hotspot is around $45°$, but probably varying within a limited direction, thus why not other VZA? Such as $40°$?

4. Is it enough to grade the uncertainty of the produced data ONLY using the number of samples? How about the pixel-based quality of ingested input MCD19A1/A3?

5. It's hard to conclude there is a "significantly but weakly" association between EVI_NAD and forest height, when $R^2=0.161$. How about the effect of terrain?

Special comments:

6. Line 59-62: The description of anisotropy definition is not appropriate. Anisotropy is the intrinsic characteristics of objects, which can be captured by directional observations, but not determined by observations. Further, the surface reflectance varies not ONLY with VZA and SZA.

7. Line 90-105: When reviewing surface reflectance products, this manuscript should not omit another important operational normalized surface reflectance products-MCD43 NBAR, and as well as the related anisotropy products-BRDF (MCD43A1 etc.), due to its acceptance in land applications all over the world.

8. Line 90: What does it mean "a factor of 3 to 10" on the accuracy?

9. Line 280: "to demonstrate the spatial and temporal distribution of…"?

10. Line 239-251: Gb, Tb? bit or Byte?? Check it through the whole text.

---

## Author Comment (AC1)

**Answer to reviewer comments**

**Manuscript:** AnisoVeg: Anisotropy and Nadir-normalized MODIS MAIAC datasets for satellite vegetation studies in South America

**Authors:** Ricardo Dalagnol, Lênio Soares Galvão, Fabien Hubert Wagner, Yhasmin Mendes Moura, Nathan Gonçalves, Yujie Wang, Alexei Lyapustin, Yan Yang, Sassan Saatchi, and Luiz Eduardo Oliveira Cruz Aragão

Reviewer comments are colored black, our answers are colored blue

**Reviewer #1 – RC1**

GENERAL COMMENTS BY REFEREE

This is an extremely useful dataset based on two decades of MODIS surface reflectance for all of South America, with interesting example applications clearly presented by the authors.

The introduction provides a succinct explanation of view-and illumination angle effects, for Modis satellite images, on the reflectance of a textured forest canopy. This problem is removed by an empirical inversion that requires several images close in time with different view and illumination angles. This is the Nadir Adjusted Reflectance (NAD) product which the authors provide, with the additional benefit of state-of-the-art MAIAC cloud removal algorithm. It is already standardized to a fixed nadir view and a fixed illumination angle, facilitating its use by a much larger number of educators and scientists.

The authors then up their game by extracting useful information from this view and illumination angle "artifact", rather than just treating it as something to be removed. This is their anisotropy (ANI) product: the difference between reflectance under a standardized back-scatter geometry and a standardized forward scatter geometry. This is like the difference between the brightness of a highly irregular textured surface photographed with the sun behind the photographer and the same surface with the photographer facing the sun. Intuitively, the difference in reflectance (or in vegetation indices) will be greater for more irregular surfaces and lesser for smoother surfaces. They show this ANI difference is useful for detecting canopy height in the Amazon, presumably because a canopy with tall trees and large crowns is more irregular than a canopy of shorter trees of similar height, that make a smoother canopy.

The paper provides three interesting examples of applications. First, they show that the Anisotropy attribute, as expressed in a single month of EVI vegetation index, distinguishes three Amazon forests which are not separable using the typical nadir Adjusted EVI. They then show that their very novel Anisotropy product is useful for estimating forest height across the entire Amazon, by comparing to GEDI lidar heights. Finally, they show that each of nine distinct leaf phenology regions of the Amazon (from an independent study) are corroborated by distinct ANI and NAD seasonal curves for the EVI vegetation Index.

ANSWERS TO REVIEWER GUIDANCE QUESTIONS IN CAPS

Are the data and methods presented new? YES

Is there any potential of the data being useful in the future? VERY HIGH

Are methods and materials described in sufficient detail? YES

Are any references/citations to other data sets or articles missing or inappropriate? NO

Is the article itself appropriate to support the publication of a data set? YES. THE ARTICLE PROVIDES EXAMPLES OF VERY USEFUL APPLICATIONS. THEIR ANISOTROPY PRODUCT WILL VERY LIKELY LEAD TO A SUITE OF NEW PAPERS ON FOREST STRUCTURE AND PHENOLOGY

Check the data quality: is the data set accessible via the given identifier? YES, I accessed the main Zenodo datasets and the two auxilliary sets. The latter allow calculating several indices based on hotspot and darkspot, that are described in Table 3 of the ESSD submission. All datasets are explained succinctly on Zenodo and in item 5 of the ESSD submission. I was also able to access the Earth Engine repository containing two Image Collections, one for anisotropy of EVI and one for Nadir-adjusted EVI. Both worked fine, using the sample code provided.

Is the data set complete? Are error estimates and sources of errors given (and discussed in the article)? Are the accuracy, calibration, processing, etc. state of the art? Are common standards used for comparison? REPLY: The authors provide the number of observations per month as a proxy for error estimation. More observations provide not only more complete data but also a more reliable BRDF inversion. The cloud masking algorithm is state-of-the-art and its originator is among the authors.

Is the data set significant – unique, useful, and complete? VERY SIGNIFICANT for scientists and educators that make use of MODIS reflectance for vegetation studies. The BRDF problem with MODIS data has been a subject of much discussion and controversy relating to Amazon forest resilience in the face of normal and extreme droughts. Here the authors not only provide corrected data, but they also turn lemons into lemonade by showing that forest structure (including canopy height) and forest leaf phenology in the Amazon are detectable by exploiting the BRDF as a measure of the anisotropic reflectance properties of canopies. So the data is also very useful. Because so much processing time is required and because few studies have previously explored the anisotropy as a useful property rather than as noise or bias, the data is unique. It is spatially complete, covering all of south America.

Consider article and data set: are there any inconsistencies within these, implausible assertions or data, or noticeable problems which would suggest the data are erroneous (or worse). If possible, apply tests (e.g. statistics). Unusual formats or other circumstances which impede such tests in your discipline may raise suspicion. NO PROBLEMS DETECTED HERE

Is the data set itself of high quality? YES

Check the presentation quality: is the data set usable in its current format and size? Are the formal metadata appropriate? THE DATA IS ACESSIBLE IN POPULAR FORMATS

Check the publication: is the length of the article appropriate? ARTICLE IS WELL WRITTEN WITH KEY EXAMPLES OF DATA APPLICATION

Is the overall structure of the article well-structured and clear? CLEAR AND CONCISE

Is the language consistent and precise? GOOD WRITING STYLE

Are mathematical formulae, symbols,abbreviations, and units correctly defined and used? YES

Are figures and tables correct and of high quality? YES.

Is the data set publication, as submitted, of high quality? YES

Reply: Thanks for your time on reading and reviewing the paper and the dataset.

LINE BY LINE COMMENTS

Lines 179-180    You obtained RTLS BRDF inversion parameters from pixels observations (having different view and solar angles) within eight day periods. What is the minimum number of pixel observations required to run the inversion in an eight-day period?

Reply: Thanks for pointing this out. It is required a minimum of three valid observations to do the inversion. We included this information on the Methods section of the updated manuscript: "A minimum of three observations in the eight-day window was required to accurately model the signal".

Lines 195-197  Are these pixel observations required for RTLS BRDF inversion conceptually identical to the "per-pixel number of samples (or observations) for each monthly composite", which is provided as ancillary data?

Reply: Yes, that is correct.

Figures 3 and 5 -- Topographic effects on ANI? In Figure 5, ANI data linearly predict forest height with $R^2 = 0.55$, presumably because the more coarsely textured surface of tall-tree canopies makes the shaded sides of trees and of large crowns occupy a greater fraction of a pixel viewed in forward scatter situation, if compared with a smooth canopy such as grassland or more even-height dicotyledon forest canopy. But your data are at 1 km resolution, so there will also be topographic irregularities within each pixel, which might also contribute to higher ANI. Have you looked into the grain and/or amplitude of topographic roughness (from SRTM) as additional explanatory variables for your scatter-plot relating ANI to forest height (Figure 5)? Do you think this will in fact be relevant?

Reply: Thanks for pointing this out. This is a good point that we missed in the original manuscript. We inspected the ANI and the SRTM data and verified similar topographic effects of higher ANI values in rough terrain. To clarify this in the manuscript, we added this paragraph below, citing a reference that showed evidence for that effect:

"$EVI_{ANI}$ results of Figures 3, 4 and 5 were affected to some extent by terrain illumination effects observed locally at some sites. For instance, topographic effects on $EVI_{ANI}$ occurred probably at the São Felix do Xingu site where topographic roughness, observed in SRTM data (results not shown), was coincident with increased $EVI_{ANI}$ values in Figure 3E. Furthermore, even in relatively flat terrains, variations in topographic aspect (surface orientation to Sun) can affect the EVI variability in MODIS data because of the different amounts of energy reflected in the NIR towards the sensor by inclined surfaces in the forward and backscattering view directions. Such effects have been observed in southern Brazil with MODIS at 250-m spatial resolution and increased in magnitude at higher spatial resolution data obtained by other sensors (Galvão et al., 2016). Therefore, it may prove useful to include topographic variables in modelling exercises to offset these effects."

Galvão, L. S., Breunig, F. M., Teles, T. S., Gaida, W., & Balbinot, R. (2016). Investigation of terrain illumination effects on vegetation indices and VI-derived phenological metrics in subtropical deciduous forests.     GIScience     and     Remote     Sensing,     53(3),     360–381. https://doi.org/10.1080/15481603.2015.1134140

In the lower three panels of Figure 3, Tapajós and Xingu Park are flat relief, so the ANI should be showing differences in canopy texture (which generally increases with canopy height in the Amazon), not topographic effects. However, I looked at the forested areas at or near the sample site in the São Felix window using SRTM and see that it is a mixture of some patches of flatter and others of more irregular relief. The ANI data there is also patchy in regions of intact forest. Is there some direct effect of topographic relief on ANI taking place in the São Felix site? Or is the patchy mosaic of low and high ANI within forest there mostly related to patchy change in canopy height/smoothness?

Reply: The reviewer is correct. This was answered in the previous answer.

Lines315-320 Fascinating. Your data is opening up new avenues for understanding leaf phenology

Reply: Thanks!

Line 315 Fix the grammar

Reply: The word 'are' was removed to fix grammar.

Line 320 change to "the central"

Reply: Corrected.

Line 350 change "on" to "for"

Reply: Corrected.

Line 355 change "consists in" to "is"

Reply: Corrected.

Line 366 change to "in the northwest"

Reply: Corrected.

Figure 6 Great figure, Distinct mutual relationships between the two indices in each pheno-region lend credence to the pheno-region classification of Xu et al (2015)

Reply: Thanks!

---

## Author Comment (AC2)

**Answer to reviewer comments**

**Manuscript:** AnisoVeg: Anisotropy and Nadir-normalized MODIS MAIAC datasets for satellite vegetation studies in South America

**Authors:** Ricardo Dalagnol, Lênio Soares Galvão, Fabien Hubert Wagner, Yhasmin Mendes Moura, Nathan Gonçalves, Yujie Wang, Alexei Lyapustin, Yan Yang, Sassan Saatchi, and Luiz Eduardo Oliveira Cruz Aragão

Reviewer comments are colored black, our answers are colored blue

**Reviewer #2 – RC2**

This study introduced the AnisoVeg product consists of monthly 1-km composites of ANI and NAD surface reflectance obtained from the MODIS over the entire South America. The paper needs a minor revision before it can be considered for publication.

1. The MODIS product MCD43 relies on multiple observations over a 16-day period, while in this study the period is extended to a month. A period of month may represent significant changes in surface especially during the vegetative stage. That would cause an inaccuracy anisotropy information of surface. Explain the possible implications of this change.

Addressed in revision: This is a good point. Land cover changes can act in two different ways, first at BRDF retrieval in MAIAC processing, and second in the compositing process. To clarify this in the text, we added two sentences:

First, in Methods section: "The MAIAC algorithm detects significant land cover changes (e.g. fire, deforestation) within the 8-day period and does not use those observations for the BRDF inversion (Lyapustin et al., 2018).".

Second, in "Time series availability and uncertainty" section, we added: "Although we use the median values to aggregate observations within months and mitigate potential land cover changes, stand-replacing changes may cause inaccurate anisotropy estimates for the given monthly estimates. Hence, we advise filtering data for land use and land cover changes before using them to obtain the most accurate anisotropy estimates."

2. To generate accurate surface anisotropy, the weight of different observations should be inconsistent in the retrieval of surface BRDF by RTLSR model. The quality and the time of the observation need to be considered together.

Addressed in revision: We expect the quality of the product is already guaranteed due to the rigorous processing of MAIAC data using time-series filtering, improved cloud and surface changes detection, and atmospheric correction to provide the best possible surface reflectance data. This could be better highlighted in the Methods section, thus we edited a sentence adding some more details about MAIAC:

"This product provides surface reflectance data corrected for atmospheric effects by the MAIAC algorithm, and controlled for cloud-free and clear-to-moderately turbid conditions with Aerosol Optical Depth (AOD) at 0.47 µm below 1.5 (Lyapustin et al., 2018). The MAIAC algorithm uses a time series approach for improved cloud filtering amongst other filters such as surface reflectance change in order to provide the most accurate surface reflectance estimates."

Regarding the time of observation, we don't understand 100% what the reviewer means, but we corrected the daily data and aggregated to monthly time scale to obtain the best possible wall-to-wall

coverage of estimates both in nadir-viewing as well as anisotropy. When we aggregate the daily data into monthly, we use the median metric to get a snapshot of the median behaviour of the estimates, that is, the more stable result within the pool of available observations. If significant changes in land cover may occur within the month, as pointed out in the previous comment, the data should still be accurate to represent most observations. If there are only a small set of observations, the user can choose not to use that pixel by filtering data out with the 'number of observations' layer.

---

## Author Comment (AC3)

**Answer to reviewer comments**

**Manuscript:** AnisoVeg: Anisotropy and Nadir-normalized MODIS MAIAC datasets for satellite vegetation studies in South America

**Authors:** Ricardo Dalagnol, Lênio Soares Galvão, Fabien Hubert Wagner, Yhasmin Mendes Moura, Nathan Gonçalves, Yujie Wang, Alexei Lyapustin, Yan Yang, Sassan Saatchi, and Luiz Eduardo Oliveira Cruz Aragão

Reviewer comments are colored black, our answers are colored blue

**Reviewer #3 – RC3**

The manuscript "AnisoVeg: Anisotropy and Nadir-normalized MODIS MAIAC datasets for satellite vegetation studies in South America" by Dalagnol et al. describes the production of a data set on vegetation anisotropy derived from MODIS data for South America. There is considerable potential for such a data set to provide useful information about the state of the land surface, and one tantalising hint that the authors provide is the result that the ANI is able to explain R2=0.55 of the variability in the GEDI canopy height signal (Fig 5., c.f. R2=0.16 for the normalised reflectance). Overall, I think this is a well written paper which describes a potentially important data set. I do have a few grumbles, mostly minor, but I hope the authors take these in the spirit in which they are intended – I am only seeking to improve the manuscript, and I am not suggesting any complex changes.

Addressed in revision: Thanks for your time to read and give comments about the manuscript. We hope to have answered most of your questions and incorporated the feedback.

Main comments:

I find it strange that MCD43 isn't mentioned anywhere. The implicit claim is, I assume, that the MODIS MAIAC data is far superior to the MOD09 and MY09 which is used to derive MCD43. I don't disagree with this, but one could also use the MCD43 data to produce similar data (not to mention that MCD43A4 contains an NBAR product which is similar in concept to the NAD in this paper). Some discussion of this in the introduction is necessary.

Addressed in revision: This is a good point and oversight from our part. We mentioned some advantages of MAIAC, including the better cloud masking and increased clear-sky observations, but did not include the traditional MODIS product name. This was fixed by including MCD43A4 name to the text and extending a bit explanation of the product's differences. The updated text now reads:

"By mitigating atmospheric interference and advancing the accuracy of surface reflectance over tropical vegetation by a factor of 3 to 10, MAIAC offers substantial improvement over conventional products such as the MOD09 (Hilker et al., 2012). Because of the better data quality retrieval, MAIAC is also an alternative to the MCD43A4 16-day Nadir Bidirectional Reflectance Distribution Function (BRDF)-Adjusted Reflectance (NBAR) product due to the less variable seasonal signal (3 to 10 times) over evergreen forests resultant from reduced effects of sun-view geometry. While the MCD43A4 NBAR product offers view-illumination correction, using the MAIAC products one can also correct for solar illumination effects at the same time."

Fig 3 does not convince me that there is complimentary information in the NAD and ANI data. Some additional metric to show how much different information is in there would be useful. For example, if

the authors calculated the principle components, how much variance would the second component explain? [note – I am convinced of this by some of the later results, but I it should ideally be demonstrated here too.]

Addressed in revision: Indeed, the figure does not quantify how complementary the information is, but show qualitatively that NAD dataset has similar values amongst the three landscapes while ANI varies from lower values at lower AGB to higher values at higher AGB. This is what we wanted to show in this part of the manuscript. As the reviewer mentioned, later on we quantify the relationship of each one to forest structure and that gives a more robust answer to this question.

L409: Table 3, captions says: "Examples of other multi-angular anisotropy indices that can be further calculated using layers of the AnisoVeg product." Initially I thought this wasn't possible as earlier the manuscript gives the impression that the layers are only ANI and NAD, however I see now this isn't the case. I suggest including a table explaining what the actual layers of the AnisoVeg product are. On a related note, although the authors call "H" the hotspot in this part of the paper, the algorithm apparently doesn't compute the value in the hotspot direction and instead use 35 degrees (see Line 217 - and I agree this is a sensible thing to do). The authors should only call this "back-scattering" so as not to give the wrong impression – it is not in the hotspot.

Addressed in revision: That is a good point. To clarify that in the manuscript, we chose not to include an additional table - because we already have too many figures and tables - but we updated the paragraph that explains what the product is with additional information:

"The AnisoVeg product consists of two main types of data spanning from 2000 to 2021 in monthly composites at 1-km spatial resolution: (a) the nadir-normalized (NAD) data; and (b) the anisotropy (ANI) data. Each data type has 10 layers corresponding to the MODIS bands 1 to 8, and two VIs (NDVI and EVI). Additionally, the product provides auxiliary layers of backward scattering and forward scattering, including part of the bands (description on section 5)."

We agree with the reviewer that the hotspot and darkspot were misleading. To clarify that, we changed the letters from H to B for backward scattering and from D to F for forward-scattering, and removed mentions to hotspot and darkspot.

Minor comments:

L59 – I think this sentence could cause confusion between what the definition of anisotropy is, and what causes it. Anisotropy is defined as the departure from Lambertian scattering, it is caused by the physical structure of media through which photons pass. I am certainly not doubting that the authors know this, but I think it could be made clearer to the reader. I am also not sure about the use of the word "mechanical" in this sentence.

Addressed in revision: Agreed. We took the liberty to use and edit upon the sentence provided by the reviewer, which is clearer than the previous one. The sentence now reads: "The anisotropy is defined as the departure from Lambertian scattering (isotropic), caused by the physical structure of media through which photons pass."

L73 – the Foody and Curran reference is a bit of an odd one to include to support this statement. Their paper doesn't really look at the influence of biophysical properties on the surface anisotropy, although it does include a correction for the influence of terrain on the observed radiance. With no disrespect to

either Foody or Curran, there are many more relevant papers that could be included here. Suggest finding some different references.

Addressed in revision: To address this concern, we replaced the reference by Sims et al. (2011) and Galvão et al. (2004) references that explored the effects of sun and view angle on vegetation indices variability:

Sims, D. A., Rahman, A. F., Vermote, E. F., & Jiang, Z. (2011). Seasonal and inter-annual variation in view angle effects on MODIS vegetation indices at three forest sites. *Remote Sensing of Environment*, *115*(12), 3112–3120. https://doi.org/10.1016/j.rse.2011.06.018

Galvao, L. S., Ponzoni, F. J., Epiphanio, J. C. N., Rudorff, B. F. T., & Formaggio, A. R. (2004). Sun and view angle effects on NDVI determination of land cover types in the Brazilian Amazon region with hyperspectral data. *International Journal of Remote Sensing*, *25*(10), 1861–1879. https://doi.org/10.1080/0143116031000159890

L110 – Again, I do not think the Foody and Curran reference is the best choice of references here. The totality of what it says on this subject is: "Terrestrial land cover surfaces are typically non-Lambertian reflectors and may exhibit a class- specific angular reflectance response. Thus data acquired at different angular geometries may help to identify and characterize land cover classes in both optical (Barnsley, 1994) and microwave (Foody, 1988) wavelengths." Whereas the current manuscript attributes "the use of multi-angular information to obtain metrics of anisotropy and extract information on forest structure" to that paper. I think this is a bit of a stretch. Suggest finding some different/additional references.

Addressed in revision: The reviewer is correct. We brought two new references to substitute the one from Foody and Curran.

Diner, D. J., Braswell, B. H., Davies, R., Gobron, N., Hu, J., Jin, Y., Kahn, R. A., Knyazikhin, Y., Loeb, N., Muller, J. P., Nolin, A. W., Pinty, B., Schaaf, C. B., Seiz, G., & Stroeve, J. (2005). The value of multiangle measurements for retrieving structurally and radiatively consistent properties of clouds, aerosols, and surfaces. *Remote Sensing of Environment*, *97*(4), 495–518. https://doi.org/10.1016/j.rse.2005.06.006

Gobron, N., Pinty, B., Verstraete, M. M., Widlowski, J. L., & Diner, D. J. (2002). Uniqueness of multiangular measurements - Part II: Joint retrieval of vegetation structure and photosynthetic activity from MISR. *IEEE Transactions on Geoscience and Remote Sensing*, *40*(7), 1574–1592. https://doi.org/10.1109/TGRS.2002.801147

L113 Whilst the Sandmeier et al., 1998 reference is appropriate here, it is most definitely not the "first" example of this type of work. It is an early example though, and perhaps that would be a better way of describing it.

Addressed in revision: The reviewer is correct. We edited the sentence to more accurately represent what Sandmeier et al 1998 have done in their study: "One of the early experiments exploring the use of anisotropy to extract information about vegetation structure were conducted by calculating the ratio between backward and forward scattering data and generating the anisotropy index (ANIX) on studying short-stature grass-type vegetation (Sandmeier et al., 1998)."

L179 Another strange reference. The Lucht and Lewis paper refenced presents a really nice results around the so-called "weights of determination" of the kernel BRDF models, but as a general reference for the RTk-LSp model it is an odd choice. A more obvious paper would be, for example, Wanner et al. (1995).

Addressed in revision: Agreed. We substituted that reference for Wanner et al., 1995.

L184: Eqn 1 – why are the labels for the kernel weights superscripted (e.g. kv) and the kernel values subscripted (e.g. Fv)? Ultimately, it doesn't greatly matter, but it would be better if these were made consistent, unless there's a good reason for not doing this.

Addressed in revision: We followed the way the subscript and superscript are organized for the parameters in Lyapustin et al 2018. We would prefer to keep them that way.

L184: Eqn 1 – I find it odd that the with kernel values are given the symbol "F" and the kernel weightsare given the symbol "k". Traditionally in the literature it has been the other way around see, for example Wanner et al. (1995) or, indeed, Lucht and Lewis (2000). This tripped me up whilst reading the paper, and a later statement appeared wrong to me due to this, so it could cause confusion. I strongly suggest changing this so that it adheres to the convention.

Addressed in revision: We can understand the confusion, but we basically followed the same convention presented in Equation 1 of Lyapustin et al 2018 and other papers related to MAIAC. Since those use the same convention, we would prefer to keep them consistent to each other, even though it deviates from the previous nomenclature from literature e.g. Wanner et al.

L200: "0.009107388 degrees" – this is quoted too precisely - 0.000000001 of a degree is a fraction of a millimetre. The text goes on to say that it is "approximately equivalent to 1 km" so really only needed to quote to that precision (say 4 or 5 d.p. in degrees).

Addressed in revision: Good point. We changed the value to 0.0091.

Typos etc:

L98 product -> products

Addressed in revision: Corrected.

L205: Here an astrix has been used as a multiplication sign, whereas in Eqn 1. an actual multiplication sign was used. Suggest making consistent.

Addressed in revision: Corrected. They are all multiplication signs now.

L315 "The EVINAD and EVIANI are seasonal variability and…" this doesn't scan. Should it say "The EVINAD and EVIANI are seasonally variable and…"?

Addressed in revision: The reviewer 1 also pointed this out. It is now corrected.

References:

Wanner, W., Li, X. and Strahler, A.H., 1995. On the derivation of kernels for kernel― driven models of bidirectional reflectance. Journal of Geophysical Research: Atmospheres. 100(D10), pp.21077-21089.

---

## Author Comment (AC4)

**Answer to reviewer comments**

**Manuscript:** AnisoVeg: Anisotropy and Nadir-normalized MODIS MAIAC datasets for satellite vegetation studies in South America

**Authors:** Ricardo Dalagnol, Lênio Soares Galvão, Fabien Hubert Wagner, Yhasmin Mendes Moura, Nathan Gonçalves, Yujie Wang, Alexei Lyapustin, Yan Yang, Sassan Saatchi, and Luiz Eduardo Oliveira Cruz Aragão

Reviewer comments are colored black, our answers are colored blue

**Reviewer #4 – RC4**

This manuscript presents the NAD (nadir-normalized) and ANI (backward-forward) surface reflectance and VI dataset produced from MAIAC MCD19A1 daily surface reflectance and 8-day MCD19A3 product. It provides possible chances to reduce uncertainty or extract information to serve the vegetation studies. Before the publish of this manuscript, further efforts may be needed to address the following general and special comments.

General comments:

1. Several anisotropy indices except ANIX have been published in the anisotropy community. There is a need to summarize out these typical indices with advantage and disadvantage, to establish the requirement of the presented metrics NAD and ANI.

Addressed in revision: Indeed, we could have explored more about this subject. We had included a brief explanation that we provide the data to calculate other anisotropic indices, but we did not explain why we chose ANI. We adjusted a paragraph in the "4. Prospective use of the dataset" section, stating that:

"Furthermore, auxiliary backward and forward scattering data are also available with the dataset. Beyond the use of the provided ANI layers, this effectively allows the computation of several other multi-angular anisotropy indices from the literature (Table 3). The advantage or disadvantage of one specific anisotropy index rather than others is not established in the literature given the range of vegetation applications and the lack of available datasets up to date. We calculated and provided only ANI due to its demonstrated relationships with Amazonian forests structure and functioning (Moura et al., 2015; Moura et al., 2016; Hilker et al., 2017). However, we expect other indices, including ratios and normalized differences between the backward and forward scattering components, offer additional possibilities for tropical vegetation studies which should be explored in future studies."

2. Confused by the "backward scattering" and "forward scattering" of Table 2, without the description of given the sun-view geometry for the fixe kernel values. Either "backward scattering" or "forward scattering"is not an sole direction.

Addressed in revision: We understand the confusion. This was due to Table 2 appeared in the text before the text explanation to backward and forward scattering, which included the solar zenith angles, view zenith angles and relative azimuth angles. To clarify, we added this information to the Table 2:

Table 2 – View-angle normalizations and corresponding BRDF kernel values.

| View-angle normalization | Solar Zenith Angle (SZA, º) | View Zenith Angle (VZA, º) | Relative Azimuth Angle (RAA, º) | $F_{0V}$ | $F_{0G}$ |
|---|---|---|---|---|---|
| Nadir | 45 | 0 | 0 | -0.04578 | - 1.10003 |
| Backward scattering | 45 | 35 | 180 | 0.22930469 | 0.017440045 |
| Forward scattering | 45 | 35 | 0 | -0.12029795 | -1.6218740 |

3. Not sure why 35°is adopted. Is it an arbitrary determination? Through hotspot is around 45°, but probably varying within a limited direction, thus why not other VZA? Such as 40°?

Addressed in revision: The 35deg VZA was justified in the original submission and followed the recommendation of a previous study from the literature according to empirical observations of the MODIS observations. This is the sentence from the manuscript: "To minimize potential errors of BRDF extrapolation, the VZA was set to 35º instead of the hotspot (45º), because 35º is a very common VZA in the empirical data distribution of the South America, and thus providing better estimates of the anisotropy (Moura et al., 2015).".

To clarify this in the updated manuscript, we edited the sentence to make it clearer: "The VZA was set to near hotspot (VZA = 35º) instead of the actual hotspot (VZA = 45º) to keep VZA closer to the actual range of MODIS observations across the South America and minimize errors coming from extrapolation of the BRDF (Moura et al., 2015)."

4. Is it enough to grade the uncertainty of the produced data ONLY using the number of samples? How about the pixel-based quality of ingested input MCD19A1/A3?

Addressed in revision: This is a good point. A complete uncertainty assessment of the MAIAC BRDF estimates and on the variability in the anisotropy estimates have been thoroughly conducted in previous studies (Hilker et al., 2012; Moura et al., 2015) – cited in the main manuscript, so we do not intend to re-do all that work by any means. One thing it would have been possible was to retrieve the per-pixel BRDF uncertainty from the daily MCD19A1 observations for every band and then get the estimate related to the median observation amongst the set of 30-day observations, and finally combine between back and forward scattering. However, this process would have been a lot of extra computation to do and would not necessarily be used by the user of the dataset. We provide the number of samples which can easily be used as a simple indicator of how robust your median 30-day observation can be - this is explained in the manuscript.

5. It's hard to conclude there is a "significantly but weakly" association between EVI_NAD and forest height, when R^2=0.161. How about the effect of terrain?

Addressed in revision: Good point. EVI is known to be affected by topography, which would appear into the $EVI_{NAD}$ estimates. Meanwhile, $EVI_{ANI}$ may have removed part of this effect by considering the difference between the two view-angles, but at the same time higher $EVI_{ANI}$ values were observed in over high slope areas in the Amazon (results not shown). This point was raised by Reviewer #1 and a paragraph was added in the previous review to address this concern regards the topography effect. To clarify this effect also for $EVI_{NAD}$, we edited the sentence to:

"Terrain illumination is a factor of spectral variability, which can affect $EVI_{NAD}$ determination and its relationship with biophysical attributes of vegetation, as shown by previous literature (Huang et al., 2010; Chen and Cao, 2012). Even at 1-km spatial resolution, $EVI_{ANI}$ results of Figures 3, 4 and 5 can be affected to some extent by terrain illumination effects observed locally at some sites. For instance, topographic effects on $EVI_{ANI}$ occurred probably at the São Felix do Xingu site where topographic roughness, observed in SRTM data (results not shown), was coincident with increased $EVI_{ANI}$ values in Figure 3E. Furthermore, even in relatively flat terrains, variations in topographic aspect (surface orientation to Sun) can affect the EVI variability in MODIS data because of the different amounts of energy reflected in the NIR towards the sensor by inclined surfaces in the forward and backscattering view directions. Such effects have been observed in southern Brazil with MODIS at 250-m spatial resolution and increased in magnitude at higher spatial resolution data obtained by other sensors (Galvão et al., 2016). Therefore, it may prove useful to include topographic variables in modelling exercises to offset these effects."

Wei, H., Zhang, L., Furumi, S., Muramatsu, K., Daigo, M., & Li, P. (2010). Topographic effects on estimating net primary productivity of green coniferous forest in complex terrain using Landsat data: A case study of Yoshino Mountain, Japan. *International Journal of Remote Sensing*, *31*(11), 2941–2957. https://doi.org/10.1080/01431160903140829

Chen, W., & Cao, C. (2012). Topographic correction-based retrieval of leaf area index in mountain areas. *Journal of Mountain Science*, *9*(2), 166–174. https://doi.org/10.1007/s11629-012-2248-2

Special comments:

6. Line 59-62: The description of anisotropy definition is not appropriate. Anisotropy is the intrinsic characteristics of objects, which can be captured by directional observations, but not determined by observations. Further, the surface reflectance varies not ONLY with VZA and SZA.

Addressed in revision: Agreed. This was also pointed out by another Reviewer. We edited the sentence to: "The anisotropy is defined as the departure from Lambertian scattering (isotropic), caused by the physical structure of media through which photons pass."

7. Line 90-105: When reviewing surface reflectance products, this manuscript should not omit another important operational normalized surface reflectance products-MCD43 NBAR, and as well as the related anisotropy products-BRDF (MCD43A1 etc.), due to its acceptance in land applications all over the world.

Addressed in revision: We agree. Another reviewer has also pointed this out. To address this concern, we edited the sentence to: "By mitigating atmospheric interference and advancing the accuracy of surface reflectance over tropical vegetation by a factor of 3 to 10, MAIAC offers substantial improvement over conventional products such as the MOD09 (Hilker et al., 2012). Because of the better data quality retrieval, MAIAC is also an alternative to the MCD43A4 16-day Nadir Bidirectional Reflectance Distribution Function (BRDF)-Adjusted Reflectance (NBAR) product due to the less variable seasonal signal (3 to 10 times) over evergreen forests resultant from reduced effects of sun-view geometry. While the MCD43A4 NBAR product offers view-illumination correction, using the MAIAC products one can also correct for solar illumination effects at the same time."

8. Line 90: What does it mean "a factor of 3 to 10" on the accuracy?

Addressed in revision: It means the MAIAC signal had 3 to 10 times less signal variability than the traditional MODIS products over evergreen forest landscapes. The MAIAC product with additional corrections and cloud filtering than the traditional products produced more stable results over time. To clarify this in the manuscript, we edited the sentence in the Introduction, as showed in the previous response.

9. Line 280: "to demonstrate the spatial and temporal distribution of…"?

Addressed in revision: We assume the sentence was not clear or there was an issue with the wording. To clarify, we edited the sentence to: "We selected three experimental areas at the Brazilian Amazon rainforests to show the spatial and temporal distribution of NAD and ANI data (rectangles in Figure 1)."

10. Line 239-251: Gb, Tb? bit or Byte?? Check it through the whole text.

Addressed in revision: We edited all occurrences of 'Gb' and 'Tb' and changed to 'GB' and 'TB' throughout the manuscript. We also added the names gigabytes and terabytes in parenthesis to clarify that in the first occurrence of each.